# An Exploration of the Labor, Financial, and Economic Factors Related to Suicide in the Republic of Kazakhstan

**DOI:** 10.3390/ijerph18136992

**Published:** 2021-06-30

**Authors:** Ken Inoue, Nursultan Seksenbayev, Nailya Chaizhunusova, Timur Moldagaliyev, Nargul Ospanova, Sholpan Tokesheva, Yersin T. Zhunussov, Nobuo Takeichi, Yoshihiro Noso, Masaharu Hoshi, Noriyuki Kawano

**Affiliations:** 1Research and Education Faculty, Medical Sciences Cluster, Health Service Center, Kochi University, Kochi 780-8520, Japan; 2The Center for Peace, Hiroshima University, Hiroshima 730-0053, Japan; mhoshi@hiroshima-u.ac.jp (M.H.); nkawano@hiroshima-u.ac.jp (N.K.); 3Department of Psychiatry, Semey Medical University, Semey 071400, Kazakhstan; nurs_7sk@inbox.ru (N.S.); timur_party@inbox.ru (T.M.); nargul_ospanova@mail.ru (N.O.); 4Department of Public Health, Semey Medical University, Semey 071400, Kazakhstan; n.nailya@mail.ru (N.C.); sholpan.man.77@mail.ru (S.T.); 5Chairman of the Board-Rector, Semey Medical University, Semey 071400, Kazakhstan; yersin.zhunussov@nao-mus.kz; 6Takeichi Clinic, Hiroshima 732-0806, Japan; htmc@topaz.ocn.ne.jp; 7Department of Health Services Management, Hiroshima International University, Hiroshima 739-2695, Japan; doc2018noso@gmail.com

**Keywords:** Republic of Kazakhstan, suicide, unemployment, risk factor, prevention

## Abstract

The Republic of Kazakhstan has one of the world’s highest suicide rates. A detailed study of the risk factors for suicide in that country is therefore important. We investigated country-wide statistics related to labor, financial, and economic factors and whether any of these factors contribute to the risk of suicide in Kazakhstan. Using the 20 year period from 2000 to 2019, we examined the annual suicide rates overall (all citizens) and for males and females in Kazakhstan, annual unemployment rates, annual rates of increase in the country’s consumer price index, annual total exports, and annual total imports. We then calculated the correlations between the suicide rates and these four items. We also performed a multiple regression analysis of the relationship between the suicide rate and those four items. The results of these analyses indicated that the unemployment rate was the correlation coefficient most highly correlated with the suicide rate; unemployment was significantly related to suicide and should be targeted as a risk factor in suicide prevention interventions in Kazakhstan. With this in mind, organizations, government agencies, and professionals in relevant fields need to devise and implement suicide prevention measures.

## 1. Introduction

The Republic of Kazakhstan is one of the republics of the former Soviet Union [1]. Kazakhstan declared independence on 16 December 1991, becoming an independent country [1]. In 2019, Kazakhstan had a population of 18.6 million [1]. Kazakhstan has one of the world’s highest overall suicide rates [2] and a particularly high suicide rate among males [3]. Only a handful of studies have been published on suicide-related issues in Kazakhstan [4,5]. Suicide is a critical social problem around the world [6,7,8,9], as it causes great sadness to the deceased’s family and friends, their acquaintances and associates, and to those who have supported them [10,11].

The conclusion of an investigation that took place over a six-year period in one area of Kazakhstan suggested a linear relationship between apparent temperatures (temperatures that people perceive) and daily suicide counts across the whole spectrum of temperatures [4]. Jumageldinov et al. noted that some cultural and historical items demonstrably related to the increase in the rate of adolescent suicides in Kazakhstan [5]. Indeed, suicide is a complex public health issue that exists at the crossroads of social science and medical science. A recent systematic review of medical and social science literature regarding the relationship between childhood maltreatment and non-suicidal self-injury confirmed a direct relationship between the two [12]. Sensory perception is another factor implicated in emotional processes and suicidality, and researchers are coming to understand that sensory processing patterns related to childhood traumatic experiences may be important in the prediction of quality of life [13].

Economic factors certainly play important roles in societal trends in suicide. Motohashi reported that the suicide rate in Japan was significantly correlated with the country’s unemployment rate [14], noting that this objective marker is an indicator of the number of people experiencing mental stress related to the ongoing economic recession and their personal economic difficulties [14]. Several studies with similar viewpoints reported a correlation between suicide and unemployment in Japan [15,16,17]. In major Australian cities, the unemployment rate was reported to be among the key factors influencing suicide [18]. Rates of manufacturing jobs and per capita income were not related to temporal variations in suicide rates within U.S. states, but they were related to differences in suicide rates among the states [19]; moreover, temporal changes in the unemployment rate were closely related to the overall U.S. suicide rate [19]. A study conducted in Ireland indicated that active labor market programs to address unemployment may play an important role in suicide prevention [20]. Interestingly, a lower unemployment rate was not necessarily related to lower suicide rates in the regions of Hong Kong and Taiwan [21]. Likewise, neither the unemployment rate nor alcohol consumption correlated with the suicide rate in a Swedish study [22]. These findings suggest cultural variation in the response to unemployment. However, the World Health Organization (WHO) has cited economic recession, including unemployment and financial loss, as a likely risk factor for suicide [23].

We reported that the suicide rate in Japan was significantly associated with the number of staff in department stores [24]. This number reflects economic trends and may serve as a simple indicator of the risk of suicide rate increase. In addition, the growth rate of the total amount of cash salaries was significantly negatively related to the suicide rates among the overall population and for males and females specifically [25], so that trends in the growth rate of the total amount in cash salaries may be useful as specific indicators of the necessity of suicide prevention measures. The value of the Nikkei Stock Average was also significantly related to the suicide rate in the overall population, as well as in males but not in females [26]. The Nikkei Stock Average (“Nikkei Index” or “Nikkei225”) is one of the representative stock price indexes of the Japanese stock market. A report based on data from the economic crisis and recession in Greece showed that interventions for suicide prevention and promotion of mental health were important for austerity-related stress [27]. One study from China utilizing the self-determination framework demonstrated that the linkage between autonomy support (as opposed to control) and lower suicide ideation was mediated by workers’ sense of meaning in life [28].

There have been few studies on factors related to suicide in Kazakhstan. Therefore, discussions of the influence on suicide of labor, financial, and economic factors in Kazakhstan can only be based on surmised, not actual, data. There have also been few international academic reports across various fields in Kazakhstan until relatively recently. Based on opinions from several countries and the WHO, we hypothesized that labor, financial, and economic factors play roles in Kazakhstan’s high suicide rate. Hence, our present study focused on suicide and several available datasets reflecting labor, financial, and economic factors in the country, and we analyzed their relative importance.

## 2. Materials and Methods

### 2.1. Data

We obtained the annual suicide rate (per population of 100,000) for the overall population and for males and females in Kazakhstan during the 20 year period from 2000 to 2019 from the WHO [29]. The annual unemployment rate (%) in Kazakhstan according to the International Monetary Fund (IMF) [30], the annual rate of increase in the consumer price index (CPI; %) [31], annual total exports (million U.S. dollars (USD)) [32], and annual total imports (million USD) [33] during the same 20 year period were obtained from Global Note. Global Note is a company (site) that specializes in official international statistical data in various fields. The unemployment rate according to the IMF was estimated from 2018 and 2019 data. All of these data were only in numerical form without individual information.

### 2.2. Statistical Analysis

We calculated the correlation coefficients between the suicide rate and (a) the unemployment rate, (b) the rate of increase in the CPI, (c) total exports, and (d) total imports throughout the study period. We also performed multiple regression analysis between the suicide rate and these four factors using Excel.

### 2.3. Interpretation of Risk Factors for Suicide

Based on the results of the statistical analysis, certain labor, financial and economic items that are risk factors for suicide in Kazakhstan were revealed, and new population-wide indicators for suicide risk were identified. We discuss specific prevention measures and planning for these risk factors.

## 3. Results

### 3.1. Annual Data for the Suicide Rate, the Unemployment Rate, the Rate of Increase in the CPI, Total Exports, and Total Imports during the 20 Year Period from 2000 to 2019

During the 20 year study period, the annual suicide rate (per population of 100,000) in the overall population of Kazakhstan varied from a minimum of 17.6 to a maximum of 39.7; the rate for males was 29.0–69.2 and that for females was 6.8–12.4 (Figure 1a,b). The highest rates were in the first portion of that period, about 10 years after independence (2000–2004), when the overall rate ranged from 38.2 to 39.7, with males at 66.4–69.2 and females at 11.5–12.4. In the period from 2005 to 2009, the overall rate ranged from 30.3 to 39.2, with males at 51.0–68.2 and females at 10.9–12.3. In the period from 2010 to 2014, the overall rate ranged from 24.2 to 29.9, with males at 40.9–50.7 and females at 8.5–10.5. Finally, in 2015–2019, the overall rate ranged from 17.6 to 23.2, with males at 29.0–39.3 and females at 6.8–8.2 (Figure 1a,b). During the same 20 year period, the annual unemployment rate (%) according to the IMF ranged from 4.78 to 12.75 (Figure 2), with ranges of 8.40–12.75 for 2000–2004, 6.55–8.11 for 2005–2009, 5.04 to 5.77 for 2010–2014, and 4.78 to 5.11 for 2015–2019. The annual rate of increase in the CPI (%) showed more variability in its trends and ranged from 5.12 to 17.15 (Figure 3), with ranges of 5.85–13.15 for 2000–2004, 7.30–17.15 for 2005–2009, 5.12 to 8.35 for 2010–2014, and 5.24 to 14.56 for 2015–2019. The annual total exports (million USD) showed more variability in its trends and ranged from USD 8639 to USD 86,449 (Figure 4), with ranges of USD 8639 to USD 20,093 for 2000–2004, USD 27,849 to USD 71,172 for 2005–2009, USD 59,971 to USD 86,449 for 2010–2014, and USD 36,685 to USD 60,956 for 2015–2019. The annual total imports (million USD) ranged from USD 5040 to USD 48,806 (Figure 5), with ranges of USD 5040 to USD 12,781 for 2000–2004, USD 17,353 to USD 37,889 for 2005–2009, USD 31,107 to USD 48,806 for 2010–2014, and USD 24,995 to USD 37,757 for 2015–2019.

### 3.2. The Correlations between the Suicide Rate and (a) the Unemployment Rate, (b) the Rate of Increase in the CPI, (c) Total Exports, and (d) Total Imports

Table 1 provides the correlation coefficients between the suicide rates in the overall population, for males, and for females and (a) the unemployment rate, (b) the rate of increase in the CPI, (c) total exports, and (d) total imports during the study period. The suicide rates in the overall population, for males, and for females were each significantly correlated with the unemployment rate, total exports, and total imports but not the rate of increase of the consumer price index. The unemployment rate especially was highly correlated with suicide rates in the overall population and for males and females.

### 3.3. The Relationships between the Suicide Rate and (a) the Unemployment Rate, (b) the Rate of Increase in the CPI, (c) Total Exports, and (d) Total Imports

The results of the multiple regression analysis of the relationship between the suicide rate and the four factors, i.e., (a) the unemployment rate, (b) the rate of increase in the CPI, (c) total exports, and (d) total imports, during the study period are indicated in Table 2a,b. Based on the results, the suicide rates of the overall population, for males, and for females in Kazakhstan were significantly associated only with the unemployment rate in the present study.

## 4. Discussion

Based on the results presented above regarding the suicide rate and the four datasets (the unemployment rate, the rate of increase in the CPI, total exports, and total imports) representing labor, financial, and economic factors, unemployment was the only significant risk factor for suicide in the overall population and among males and females in Kazakhstan. Our literature search yielded several studies conducted in developed countries that obtained similar results [14,15,16,17,18,19,20,22,34]. Another study reported that suicides increase especially during periods of severe economic crisis [35]. Widespread economic downturn and unemployment cause personal hardships; if those hardships continue, they can lead to mental exhaustion. In other words, individuals are subjected to excessive mental stress that could lead to suicide.

In Kazakhstan, which is still a developing country, unemployment causes hardships and intense mental and physical anguish that can lead to suicide, as in developed countries.

Japan had a large, abrupt increase in suicides about 20 years ago [14]. The council examined suicide prevention measures on a prefectural and local municipal basis and subsequently proposed suicide prevention measures addressing economic and life problems for the country as a whole and in individual prefectures, major metropolitan areas, and local municipalities based on the circumstances specific to those regions, which it then implemented [36].

In Kazakhstan, preventive measures addressing labor, financial, and economic issues, such as economic and life problems, are needed; medical support for the mental and physical states that result from those problems must be provided, and a government response to those mental and physical states is important. A network of close collaborations among personnel in these areas is needed, and a practical response manual could be created and followed to provide suicide prevention guidelines. Like Japan, Kazakhstan needs a country-specific approach to preventing suicides due to economic and subsequent life problems. Measures and programs that have been effective in other countries can be studied and imitated when appropriate. Actions that draw upon Kazakhstani culture should be devised, and the advantages of collaboration among professionals in multiple fields could be utilized. Unemployment is clearly a particular risk for suicide in many countries, but not in others; it would be interesting to consider what factors distinguish the two categories. It is possible that specific measures that have been effective in preventing suicide due to unemployment in other countries would work in Kazakhstan. Systems of collaboration to prevent suicide are also needed.

At present, COVID-19 is a major problem around the world, including in Kazakhstan. Worsening mental states (e.g., depression and anxiety) due to COVID-19, mental issues due to the economic impacts of the pandemic, and pessimism brought about by the disease have garnered attention as COVID-19 continues to spread around the world [37,38]. Among these specific effects, unemployment may be the greatest risk factor for suicide. Like other countries, Kazakhstan locked down the entire country with the continued transmission of COVID-19, and individual regions of the country responded by locking down or instituting self-isolation [39]. The likely economic impacts of these changes [40] will continue to reverberate for several years. It is necessary to perform further research on suicide in Kazakhstan. Suicide prevention is important from a young age, and prevention programs have improved the ability to recognize and to verbally convey emotions while maintaining stable initial characteristics, such as psychological well-being and positive expectations for the future [41].

There are some limitations to this study. The results were based on the relationships between the suicide rate and four factors and did not define causality. Furthermore, the present study was a discussion about economic-based risk factors for suicide based on statistical analysis of only numerical data without individual information. Analyses of case-based studies of suicide would be important from the viewpoint of clinical and societal interventions. The study did not examine differences between Soviet-era and post-Soviet suicide rates, so it cannot shed light on the effect of a command economy vs. a market economy on suicide. The study did not probe the stark difference between male and female suicide rates in Kazakhstan or compare them to those of other countries and cultures where masculinity may be defined differently. The study did not examine differences between conditions in countries like Japan and the U.S., where unemployment is related to suicide risk, and in Taiwan and Sweden, where it is not. It did not elaborate on what types of prevention programs have worked or examine the data for clues to whether those methods might be applicable to Kazakhstan. However, it is a timely exploration given the current economic conditions worldwide and it opens up these issues as opportunities for further study among researchers in a variety of academic fields.

## 5. Conclusions

Our analyses revealed that, among several labor, financial, and economic factors, unemployment is a major risk factor for suicide in the overall population and for males and females in Kazakhstan. As prevention measures, it is important for labor, financial, and economic divisions of government at both the local and national level to collaborate in planning for economic downturns and to provide stabilizing supports such as jobs programs or organized volunteerism to maintain individuals’ sense of self and purpose. In addition, extra efforts to implement suicide awareness and prevention programs at schools, community centers, and places of worship during times of economic hardship are called for.

## Figures and Tables

**Figure 1 ijerph-18-06992-f001:**
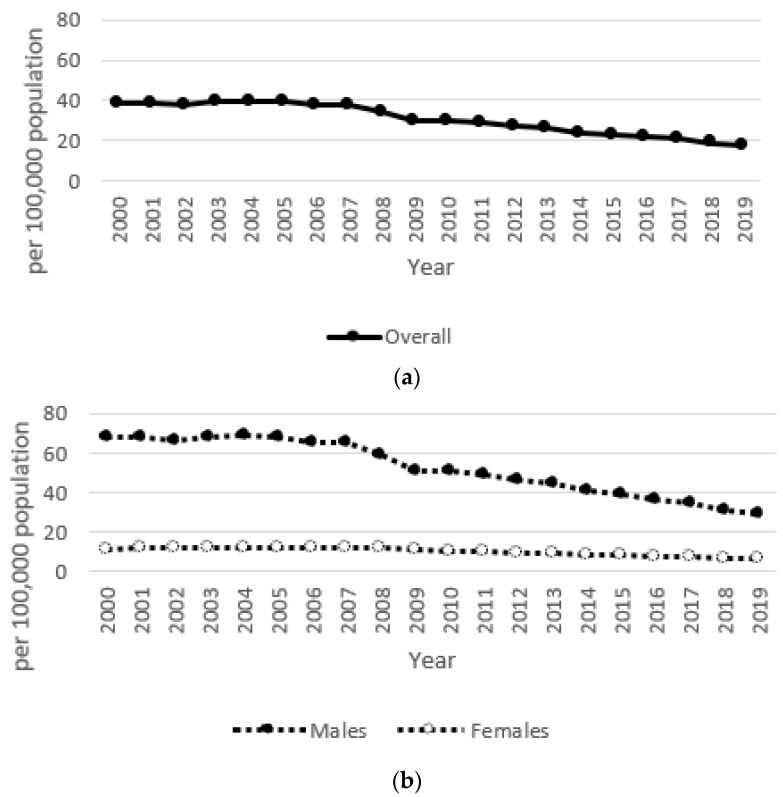
(**a**)The annual suicide rate in the overall population of Kazakhstan during the 20 year period from 2000 to 2019. (**b**)The annual suicide rates for males and females in Kazakhstan during the 20 year period from 2000 to 2019.

**Figure 2 ijerph-18-06992-f002:**
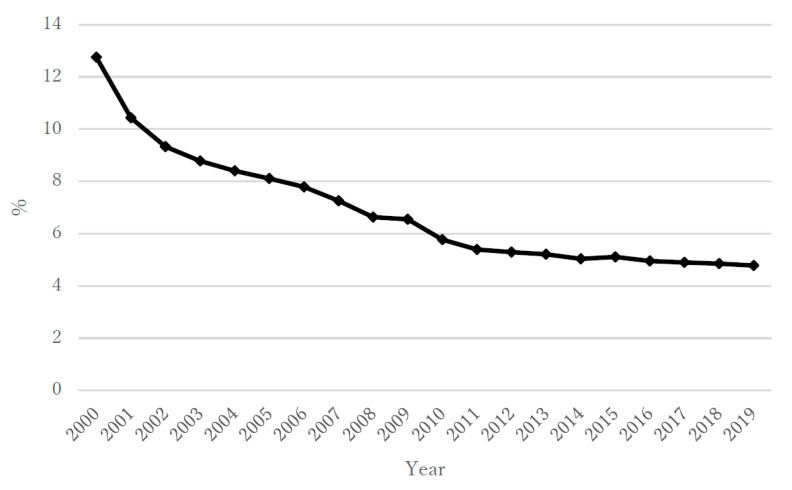
The annual unemployment rate in Kazakhstan during 2000–2019.

**Figure 3 ijerph-18-06992-f003:**
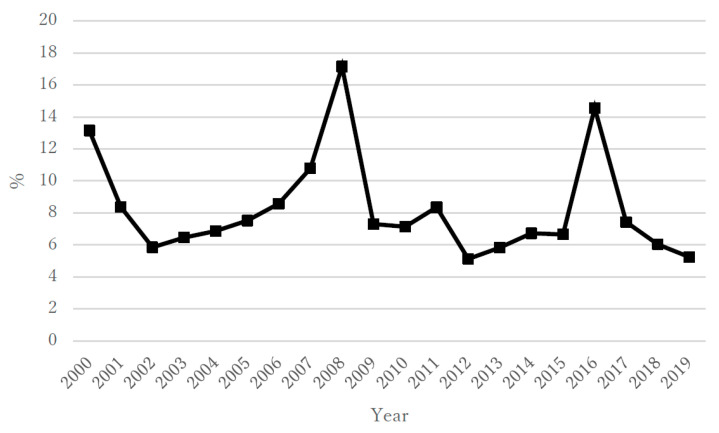
The annual rate of increase in the consumer price index (CPI) in Kazakhstan during 2000–2019.

**Figure 4 ijerph-18-06992-f004:**
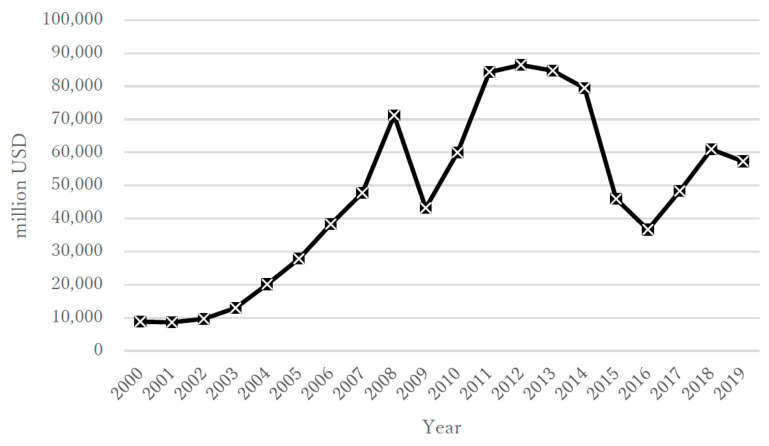
Annual total exports in Kazakhstan during 2000–2019.

**Figure 5 ijerph-18-06992-f005:**
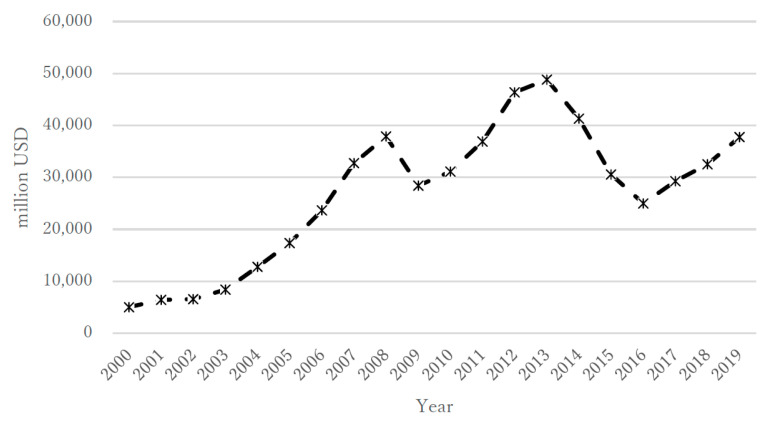
Annual total imports in Kazakhstan during 2000–2019.

**Table 1 ijerph-18-06992-t001:** Correlations between the suicide rate and each of the four factors.

	Overall	Males	Females
Unemployment rate	r = 0.833, *p* < 0.001	r = 0.846, *p* < 0.001	r = 0.742, *p* < 0.001
Rate of increase in the consumer price index	r = 0.217, *p* = 0.358	r = 0.214, *p* = 0.365	r = 0.233, *p* = 0.323
Total exports	r = −0.602, *p* = 0.005	r = −0.618, *p* = 0.004	r = −0.503, *p* = 0.024
Total imports	r = −0.658, *p* = 0.002	r = −0.673, *p* = 0.001	r = −0.558, *p* = 0.011

**Table 2 ijerph-18-06992-t002:** (**a**) Multiple regression analysis was performed to determine the relationships among the suicide rate and the four factors: (1) the unemployment rate, (2) the rate of increase in the CPI, (3) total exports, and (4) total imports. (**b**) Regression lines in the multiple regression analysis.

	Overall	Males	Females
Adjusted R^2^	R^2^ = 0.624	R^2^ = 0.648	R^2^^ ^ = 0.456
Unemployment rate (X1)	*p* = 0.004	*p* = 0.003	*p* = 0.016
Rate of increase in the consumer price index (X2)	*p* = 0.954	*p* = 0.980	*p* = 0.813
Total exports (X3)	*p* = 0.719	*p* = 0.727	*p* = 0.714
Total imports (X4)	*p* = 0.916	*p* = 0.910	*p* = 0.950
(**a**)			
Overall	y = 3.3067X_1_ + 0.0208X_2_ + 6.2101 × 10^−5^X_3_ − 4.0516 × 10^−5^X_4_ + 6.015		
Males	y = 6.0240X_1_ + 0.0162X_2_ + 0.0001X_3_ − 7.5825×10^−5^X_4_ + 8.235		
Females	y = 0.8096X_1_ + 0.0263X_2_ + 1.9521×10^-5^X_3_ − 7.4097 × 10^−6^X_4_ + 3.779		
(**b**)			

Response variable (Y): the suicide rates. Explanatory variable (X): the unemployment rate (X_1_), the rate of increase in the CPI (X_2_), total exports (X_3_), and total imports (X_4_).

## Data Availability

Not applicable.

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
