# Peer review of "An Exploration of the Labor, Financial, and Economic Factors Related to Suicide in the Republic of Kazakhstan"

_ijerph, 2021, doi:10.3390/ijerph18136992_

Round 1

Reviewer 1 Report

Thank you for the opportunity to revise the paper entitled “An exploration of the labor, financial, and economic factors 2 that are particularly related to suicide in the Republic of Kazakhstan”.

The introduction lacks a paragraph with the relative bibliography dedicated to the phenomenon of suicide in general and to the importance of monitoring the trend of suicide rates in this country. The introduction in general should be made simpler, the bibliography is good but the description of the studies in the introduction is excessive and should be simplified. Studies on suicide in Japan must only be mentioned and not developed in detail. Furthermore, in the final part of the introduction, the working hypothesis is completely missing and above all the objectives must be clearly postulated.

In the discussion section the paragraph on Japan should be revised as the results and scope of the study does not involve Japan.

The paragraph on the limitations of study is completely missing. Limitations are an important part of every scientific research.

In the conclusion authors the authors should better specify the practical implications of the study and which prevention strategies should be introduced to limit the phenomenon.

Author Response

Reviewer 1:

Thank you for the opportunity to revise the paper entitled “An exploration of the labor, financial, and economic factors 2 that are particularly related to suicide in the Republic of Kazakhstan”.

→Thank you for your polite comment. We appreciate this opportunity to you.

The introduction lacks a paragraph with the relative bibliography dedicated to the phenomenon of suicide in general and to the importance of monitoring the trend of suicide rates in this country. The introduction in general should be made simpler, the bibliography is good but the description of the studies in the introduction is excessive and should be simplified. Studies on suicide in Japan must only be mentioned and not developed in detail. Furthermore, in the final part of the introduction, the working hypothesis is completely missing and above all the objectives must be clearly postulated.

→Thank you for your valuable comments. We thought your comments carefully, and we added as follows.

・Page 1, line 27-29: Suicide is a critical social problem around the world [6-9], as it causes great sadness to the person's family and friends, their acquaintances and associates, and those who have supported them [10,11].

・Page 2, line 69-74: A report based on data of economic crisis and recession in Greece showed that interventions for suicide prevention and promotion of mental health were important for austerity-related stress [27]. One study from China utilizing the self-determination framework demonstrated that the linkage between autonomy support (as opposed to control) and lower suicide ideation was mediated by workers' sense of meaning in life [28].

・Page 2, line 78-80: Based on opinions from several countries and the WHO, we believed that labor, financial, and economic factors play roles in Kazakhstan’s high suicide rate. Hence, our・・・.

In the discussion section the paragraph on Japan should be revised as the results and scope of the study does not involve Japan.

→Thank you for your advice. Sorry. We delated almost parts in previous 4.Discussion. Suggestion parts from other reviewer was only remaining.

The paragraph on the limitations of study is completely missing. Limitations are an important part of every scientific research.

→Thank you for your precious advice. Reviewer 3 was also shown description of limitations. We added as follows.

・Page 8, line 211-226: There are some limitations to this study. The results were based on the relationships between the suicide rate and four factors and do not define causality. Furthermore, the present study is a discussion about economic-based risk factors for suicide based on statistical analysis from only numerical data without individual information. Analyses of case-based studies of suicide would be important from the viewpoint of clinical and societal interventions. The study does not examine differences between Soviet-era and post-Soviet suicide rates, so it cannot shed light on the effect of a command economy vs market economy on suicide. The study does not probe the stark difference between male and female suicide rates in Kazakhstan or compare them to those of other countries and cultures where masculinity may be defined differently. The study does not examine differences between conditions in countries like Japan and the U.S., where unemployment is related, and in Taiwan and Sweden, where it is not related. It does not elaborate what types of prevention programs have worked or examine the data for clues to whether those methods might be applicable to Kazakhstan. However, it is a timely exploration given the current economic conditions worldwide and opens up these issues as opportunities for further study among researchers in a variety of academic fields.

In the conclusion authors the authors should better specify the practical implications of the study and which prevention strategies should be introduced to limit the phenomenon.

→Thank you for your valuable suggestion. We added as follows.

・Page 8, line 231-236: As prevention measures, it is important for labor, financial, and economic divisions of government at both the local and national level to collaborate in planning for economic downturns and provide stabilizing supports such as jobs programs or organized volunteerism to maintain individuals’ sense of self and purpose. In addition, extra efforts to implement suicide awareness and prevention programs at schools, community centers and places of worship during times of economic hardship are called for.

Other

→・Reference No is new in this time.

・A profession editing service (KN International) edited all of this revise manuscript.

Very sincerely yours,

Correspondence: Ken Inoue, MD, PhD

Research and Education Faculty, Medical Sciences Cluster, Health Service Center, Kochi University

The Center for Peace, Hiroshima University

1-1-89, Higashisendamachi, Naka-ku, Hiroshima-shi, Hiroshima 730-0053, Japan

2-5-1, Akebono-cho, Kochi-shi, Kochi 780-8520, Japan

Tel: +81-88-844-8158, Fax: +81-88-844-8089

E-mail address: [email protected] (K. Inoue)

https://orcid.org/0000-0002-0710-481X

Reviewer 2 Report

Abstract           

The present article is aimed at investigating the relationship between suicide in the Republic of Kazakhstan and labor, financial and economic factors. The study considered annual suicide rates in the period 2000 and 2019, for both females and males. The results of this study indicate that unemployment is significantly related to suicide and should therefore be targeted as a risk factor in suicide prevention interventions in Kazakhstan.

The “abstract” session conforms with the journal guidelines, stating – under the “front matter” section- that “The abstract should be a total of about 200 words maximum”. Overall, the abstract seems clear and able to give a general preview of the content of the article.

Keywords

As far as the keywords are concerned, they all seem appropriate except for the word “problems”, which seems to be too generic, thus not helping readers select this article.

  1. Introduction

Overall, the introductory session requires a major revision. Some parts of the introductory session contain non-relevant information for the purpose of understanding the background of the present research. Moreover, the literature presented is biased toward Japan and a broader spectrum on the international literature on suicide and suicide prevention should be provided. Refences 1 and 5 are overly cited, for example, so it is advisable for the authors to expand their bibliography and look further into the literature.

It is not very clear why the authors focus on suicide data regarding a country like Japan, which has a very different economic and social background from Kazakhstan. Therefore, the authors are invited to either motivate this comparison or to refer to international literature more generally.

A detailed description of the required amendments is provided as follows:

- Page 1, line 23: It would be interesting to have direct information from the Ministry of Foreign Affairs of Kazakhstan, instead of that of Japan – see reference 1.

- Page 1, lines 23-30: The authors provide readers with some non-relevant pieces of information about the history of Kazakhstan. They are invited to keep only those parts that are necessary for readers to understand the background and the rationale for this research. Therefore, the information relative to the capital of Kazakhstan or the extension of its territory are not relevant for the just mentioned purpose. In case they were relevant, the authors should make this relevance explicit.

- Page 1, lines 32-34: This passage should be re-phrased in order to be clearer. More specifically, the authors are invited to explain the expression “apparent temperatures”.

- Page 1, lines 37-43: The introductory session should present the findings of previous research in a discursive way and no technical and statistical measurements should be provided in this session. Moreover, the authors could clarify the expression “SAVA syndemic continuum”.

- Page 1, lines 34, 43 and 46: It would be better for the authors to use the correct form of “et. al” in in-text citations.

- Page 2, lines 64-65: It would be useful to have a connection to express that contrary to what has been found in the just cited countries, Sweden and Hong Kong and Taiwan show different, opposing results. Moreover, in this whole paragraph, as in the rest of the introductory session, a more general reference on findings of international relevance, from, for instance, the World Health Organization, should be provided.

- Page 2, lines 82-84: The authors are invited to clarify the first two lines. Moreover, more general information might be useful in this part.

- Page 2, line 88: The authors are invited to briefly explain what “Nikkei Stock Average” is.

  1. Materials and Methods

The present section needs major revisions. More specifically, the authors should expand on the statistical analysis performed in this research.

In the section 2.1, it needs to be explained what the “GLOBAL NOTE” is. Section 2.3 doesn’t add value to the text, so it should be further explained and expanded or directly removed.

  1. Results

Overall, the findings of the study are presented in a clear way and the visual information helps in the understanding of the discursive presentation of the results, but some parts should be revised and explained better.

 Some suggested revisions are:

- Page 3, section 3.1: This section should be revised, since is not very clear the distribution of the minimum and the maximum values of the variables in time. For example, it’s written that “the annual suicide rate in overall population of Kazakhstan varied from a min. of 17.6 to a max. of 39.7”, while in Figure 1 it’s shown a downward trend for the annual suicide rate. So, it should be better explained the fact that these are the extreme of the values occurred during the 20-year period.

- Page 3, figure 1: it is advisable to use a different symbol for either the “overall population” or the “male population”, since the two symbols are very similar, and it might be difficult for the reader to distinguish between the two.

- Page 6, table 1: the “p value” should be added to the correlation table

- Page 6, section 3.3: this section of the manuscript should be revised, since it’s not very clear the procedure used and the meaning of all the values presented.  Also, a table presenting the results of the multiple regression analysis could be integrate this paragraph, in order to help the reader to observe the important associations between the values.

  1. Discussion

The discussion session requires revision. As in the introductory session, some non-relevant passages are present in this session. For instance, paragraphs number two (lines 181-190) and the first part of paragraph three (lines 191-193), contain information which might not be necessary for readers to understand the context of this research or, their relevance should be made explicit. In paragraph number four (197-204) the reflection could be broadened regarding the likelihood that the incidence of COVID-19 has increased suicide rates, by citing the work that has been previously done on the psychological impact of the pandemic and the economic crisis in some sectors. Moreover, it’s important to take into account the pandemic situation in the implementation of suicide prevention programs.

In line 205, is stated “Japan had a large, abrupt increase in suicide rates about 20 years ago” but there is no refence cited. Also, at the end of the fifth paragraph, between line 211 and 216, there is no refence cited, therefore the authors need to add the proper literature to validate their statement.

Another point that should be discussed is that between the suicide rate and unemployment was found a correlation, which means that there is a mutual relationship between the two variables. But that doesn’t necessarily define causality, so this aspect could be presented in the discussion, and it could also be considered the existence of third mediating variables.

  1. Conclusions

The conclusion part should be further expanded, so the authors are invited to present identified limitations for the present research and possible insights for future research.

References

This session isn’t entirely in line with the formatting required by the journal guidelines, since the number volume of the journals should be in written in italics.  Overall, the authors are invited to make the reference more diverse in terms of sources, as a lot of the literature focuses on Japan and on literature produced by the same author (see, for instance, 7, 12, 13, 19-27). As regards suicide prevention interventions, some suggested literature the authors could refer to is:

- Testoni, I., Tronca, E., Biancalani, G., Ronconi, L., & Calapai, G. (2020). Beyond the wall: death education at middle school as suicide prevention. International journal of environmental research and public health17(7), 2398. doi: 10.3390/ijerph17072398

Use of English

Overall, the authors use a correct and understandable English language. Some parts of the text should be further explained, as previously mentioned above.

Overview

In general, the present research requires major revisions. The authors are invited to re-structure both the introduction and discussion sessions. They are invited to maintain only those pieces of information relevant to understand the background and the context in which this research is located. Moreover, more international literature should be used in the references for this article, and it should be avoided to use too many references from the same author.

Author Response

Reviewer 2:

Abstract           

The present article is aimed at investigating the relationship between suicide in the Republic of Kazakhstan and labor, financial and economic factors. The study considered annual suicide rates in the period 2000 and 2019, for both females and males. The results of this study indicate that unemployment is significantly related to suicide and should therefore be targeted as a risk factor in suicide prevention interventions in Kazakhstan.

→Thank you for your advice. We corrected as follows.

・Page 1, line 14-16: The results of these analyses indicated that the unemployment rate was the correlation coefficients most highly correlated with the suicide rate; unemployment was significantly related to suicide and should be targeted as a risk factor in suicide prevention interventions in Kazakhstan.

The “abstract” session conforms with the journal guidelines, stating – under the “front matter” section- that “The abstract should be a total of about 200 words maximum”. Overall, the abstract seems clear and able to give a general preview of the content of the article.

→Thank you for your comment. We corrected as follows.

・Abstract in the present is 185 words.

Keywords

 As far as the keywords are concerned, they all seem appropriate except for the word “problems”, which seems to be too generic, thus not helping readers select this article.

 →Thank you for your advice. We changed from “problems” to “prevention”.

  1. Introduction

 Overall, the introductory session requires a major revision. Some parts of the introductory session contain non-relevant information for the purpose of understanding the background of the present research. Moreover, the literature presented is biased toward Japan and a broader spectrum on the international literature on suicide and suicide prevention should be provided. Refences 1 and 5 are overly cited, for example, so it is advisable for the authors to expand their bibliography and look further into the literature.

It is not very clear why the authors focus on suicide data regarding a country like Japan, which has a very different economic and social background from Kazakhstan. Therefore, the authors are invited to either motivate this comparison or to refer to international literature more generally.

 A detailed description of the required amendments is provided as follows:

- Page 1, line 23: It would be interesting to have direct information from the Ministry of Foreign Affairs of Kazakhstan, instead of that of Japan – see reference 1.

→Thank you for your comment. We are glad.

- Page 1, lines 23-30: The authors provide readers with some non-relevant pieces of information about the history of Kazakhstan. They are invited to keep only those parts that are necessary for readers to understand the background and the rationale for this research. Therefore, the information relative to the capital of Kazakhstan or the extension of its territory are not relevant for the just mentioned purpose. In case they were relevant, the authors should make this relevance explicit.

→Thank you for your suggestion. We delated the parts.

- Page 1, lines 32-34: This passage should be re-phrased in order to be clearer. More specifically, the authors are invited to explain the expression “apparent temperatures”.

→Thank you for your advice. We added as follows.

・Page 1, line 31-32: (temperatures that people perceive)

- Page 1, lines 37-43: The introductory session should present the findings of previous research in a discursive way and no technical and statistical measurements should be provided in this session. Moreover, the authors could clarify the expression “SAVA syndemic continuum”.

→Thank you for your advice. We delated the parts.  According to suggestion of Reviewer 3, we delated contents of [previous Ref No5], and added ‘A recent systematic review of medical and social science literature regarding the relationship between childhood maltreatment and non-suicidal self-injury confirmed a direct relationship between the two [12]. Sensory perception is another factor implicated in emotional processes and suicidality, and researchers are coming to understand that sensory processing patterns related to childhood traumatic experiences may be important in the prediction of quality of life [13].’ in Page1, line 36-41.

- Page 1, lines 34, 43 and 46: It would be better for the authors to use the correct form of “et. al” in in-text citations.

→Thank you for your advice. We corrected the parts “et. al”.

- Page 2, lines 64-65: It would be useful to have a connection to express that contrary to what has been found in the just cited countries, Sweden and Hong Kong and Taiwan show different, opposing results. Moreover, in this whole paragraph, as in the rest of the introductory session, a more general reference on findings of international relevance, from, for instance, the World Health Organization, should be provided.

→Thank you for your suggestion. We corrected them as follows.

・Page 2, line 54-55: Interestingly, a lower unemployment rate was not necessarily related to lower suicide rates in the regions of Hong Kong and Taiwan [21].

・Page 2, line 58-59: Meanwhile, the World Health Organization (WHO) did cite economic recession including unemployment and financial loss as a likely risk factor for suicide [23].

- Page 2, lines 82-84: The authors are invited to clarify the first two lines. Moreover, more general information might be useful in this part.

→Thank you for your specific advice. We added as follows.

・Page 2, line 61-62: This number reflects economic trends and may serve as a simple indicator of the risk of suicide rate increase.

・Page 2, line 64-66: so that trends of the growth rate of total amount in cash salaries may be useful as a specific indicator of the necessity of suicide prevention measures.

- Page 2, line 88: The authors are invited to briefly explain what “Nikkei Stock Average” is.

→Thank you for your specific suggestion. We added as follows.

・Page 2, line 67-69: Nikkei Stock Average (“Nikkei Index” or “Nikkei225”) is one of the representative stock price indexes of the Japanese stock market.

  1. Materials and Methods

 The present section needs major revisions. More specifically, the authors should expand on the statistical analysis performed in this research.

In the section 2.1, it needs to be explained what the “GLOBAL NOTE” is. Section 2.3 doesn’t add value to the text, so it should be further explained and expanded or directly removed.

→Thank you for your valuable comments.

We added as follows.

・Page 2, line 91-92 in Section 2.1: We described that ‘GLOBAL NOTE is a company (site) that specializes in official international statistical data in various fields.’.

We added further specific prevention contents about the risk factor for suicide according to the advice from Reviewer 1, therefore we described as follows.

・Page 3, line 108-109 in Section 2.3: We discuss specific prevention measures and planning for these risk factors.

  1. Results

 Overall, the findings of the study are presented in a clear way and the visual information helps in the understanding of the discursive presentation of the results, but some parts should be revised and explained better.

 Some suggested revisions are:

- Page 3, section 3.1: This section should be revised, since is not very clear the distribution of the minimum and the maximum values of the variables in time. For example, it’s written that “the annual suicide rate in overall population of Kazakhstan varied from a min. of 17.6 to a max. of 39.7”, while in Figure 1 it’s shown a downward trend for the annual suicide rate. So, it should be better explained the fact that these are the extreme of the values occurred during the 20-year period.

- Page 3, figure 1: it is advisable to use a different symbol for either the “overall population” or the “male population”, since the two symbols are very similar, and it might be difficult for the reader to distinguish between the two.

→Thank you for your precious suggestions. We added as follows.

・Page 3, line 108-128: During the 20-year study period, the annual suicide rate (per 100,000 population) in the overall population of Kazakhstan varied from a minimum of 17.6 to a maximum of 39.7; the rate in males was 29.0–69.2, and that in females was 6.8–12.4 (Figures 1a, 1b). The highest rates were in the first portion of that period, about 10 years after independence (2000-2004), when the overall rate ranged from 38.2 to 39.7, with males at 66.4–69.2 and females 11.5–12.4. In the period 2005-2009, the overall rate ranged from 30.3 to 39.2, with males at 51.0–68.2 and females 10.9–12.3. In the period 2010-2014, the overall rate ranged from 24.2 to 29.9 with males at 40.9–50.7 and females 8.5–10.5. Finally, in 2015-2019, the overall rate ranged from 17.6 to 23.2 with males at 29.0–39.3 and females 6.8–8.2 (Figures 1a,1b). During the same 20-year period, the annual unemployment rate (%) according to the IMF ranged from 4.78 to 12.75 (Figure 2), with ranges of 8.40-12.75 for 2000-2004, 6.55-8.11 for 2005-2009, 5.04 to 5.77 for 2010-2014 and 4.78 to 5.11 for 2015-2019. The annual rate of increase in the CPI (%) showed more variability in trends and ranged from 5.12 to 17.15 (Figure 3), with ranges of 5.85-13.15 for 2000-2004, 7.30-17.15 for 2005-2009, 5.12 to 8.35 for 2010-2014 and 5.24 to 14.56 for 2015-2019. The annual total exports (million USD) showed more variability in trends and ranged from $8,639 to $86,449 (Figure 4), with ranges of $8,639-$20,093 for 2000-2004, $27,849-$71,172 for 2005-2009, $59,971 to $86,449 for 2010-2014 and $36,685 to $60,956 for 2015-2019. The annual total imports (million USD) ranged from $5,040 to $48,806 (Figure 5), with ranges of $5,040-$12,781 for 2000-2004, $17,353-$37,889 for 2005-2009, $31,107 to $48,806 for 2010-2014 and $24,995 to $37,757 for 2015-2019.

→We separated Figure1 into 1a (overall) and 1b (males and females). Probably we thought that it's easier to understand for the reader.

・Page 3-4, line 130 and 132: Figure 1a. The annual suicide rate in overall population of Kazakhstan during the 20-year period 2000–2019.’, and ‘Figure 1b. The annual suicide rate in males and females of Kazakhstan during the 20-year period 2000–2019.’

- Page 6, table 1: the “p value” should be added to the correlation table

→Thank you for your specific suggestion. We added “p value” in Page 6, table 1 as follows.

Table 1. Correlation between the suicide rate and four factors.

Overall

Males

Females

Unemployment rate

r=0.833, p=5.166E-06

r=0.846, p=2.672E-06

r=0.742, p=2E-04

Rate of increase in the consumer price index

r=0.217, p=0.358

r=0.214, p=0.365

r=0.233, p=0.323

Total exports

r=-0.602, p=0.005

r=-0.618, p=0.004

r=-0.503, p=0.024

Total imports

r=-0.658, p=0.002

r=-0.673, p=0.001

r=-0.558, p=0.011

Based on the results, we corrected as follows.

・Page 6, line 145-149: The suicide rates in the overall population, males, and females were each significantly correlated with the unemployment rate, total exports and total imports but not the rate of increase of the consumer price index. Especially, the unemployment rate was highly correlated with suicide rates in the overall population, males, and females.

- Page 6, section 3.3: this section of the manuscript should be revised, since it’s not very clear the procedure used and the meaning of all the values presented.  Also, a table presenting the results of the multiple regression analysis could be integrate this paragraph, in order to help the reader to observe the important associations between the values.

→Thank you for your specific suggestion and advice. We added Tables 2a and 2b.

Table 2a. Multiple regression analysis was performed to determine the relationships among the suicide rate and the four factors. (a) the unemployment rate, (b) the rate of increase in the CPI, (c) total exports, and (d) total imports.

Overall

Males

Females

Adjusted R2

R2= 0.624

R2= 0.648

R2= 0.456

Unemployment rate (X1)

p=0.004

p=0.003

p=0.016

Rate of increase in the consumer price index (X2)

p=0.954

p=0.980

p=0.813

Total exports (X3)

p=0.719

p=0.727

p=0.714

Total imports (X4)

p=0.916

p=0.910

 p=0.950

Table 2b. Regression lines in the multiple regression analysis.

Overall

y = 3.3067X1 + 0.0208X2 + 6.2101E-05X3 − 4.0516E-05X4 + 6.015

Males

y = 6.0240X1 + 0.0162X2 + 0.0001X3 − 7.5825E-05X4 + 8.235

Females

 y = 0.8096X1 + 0.0263X2 + 1.9521E-05X3 − 7.4097E-06X4 + 3.779

Response variable (Y): The suicide rates. Explanatory variable (X): the unemployment rate (X1), the rate of increase in the CPI (X2), total exports (X3), and total imports (X4).

→We corrected as follows.

・Page 6, line 153-156: The results of the multiple regression analysis of the relationship between the suicide rate and the four factors, i.e., (a) the unemployment rate, (b) the rate of increase in the CPI, (c) total exports, and (d) total imports during the study period are indicated in Tables 2a and 2b.

・Page 6, line 156-158: Based on the results, the suicide rates of the overall population, males, and females in Kazakhstan were significantly associated only with the unemployment rate in the present study.

  1. Discussion

 The discussion session requires revision. As in the introductory session, some non-relevant passages are present in this session. For instance, paragraphs number two (lines 181-190) and the first part of paragraph three (lines 191-193), contain information which might not be necessary for readers to understand the context of this research or, their relevance should be made explicit. In paragraph number four (197-204) the reflection could be broadened regarding the likelihood that the incidence of COVID-19 has increased suicide rates, by citing the work that has been previously done on the psychological impact of the pandemic and the economic crisis in some sectors. Moreover, it’s important to take into account the pandemic situation in the implementation of suicide prevention programs.

→Thank you for your specific suggestion and advice. We delated their almost parts

Based on your advice, we delated their almost parts. COVID-19 -related item was used for future perspectives in Discussion.

In line 205, is stated “Japan had a large, abrupt increase in suicide rates about 20 years ago” but there is no refence cited. Also, at the end of the fifth paragraph, between line 211 and 216, there is no refence cited, therefore the authors need to add the proper literature to validate their statement.

→Thank you for your suggestion. We added as follows.

・Page 7, line 177-182: Japan had a large, abrupt increase in suicides about 20 years ago [14]. The Council examined suicide prevention measures on a prefectural and local municipal basis and subsequently proposed suicide prevention measures addressing economic and life problems for the country as a whole, and in individual prefectures, major metropolitan areas, and local municipalities based on the circumstances specific to those regions, which it then implemented [36].

・36. Takahashi, Y.; Takahashi, S.; Imamura, Y.; Yamashita, R. The national strategies for suicide prevention by the United Nation/World Health Organization and the present situation of suicide in the East Asia. Seishin Shinkeigaku Zasshi 2014, 116, 690-696 (in Japanese, Abstract in English).

Another point that should be discussed is that between the suicide rate and unemployment was found a correlation, which means that there is a mutual relationship between the two variables. But that doesn’t necessarily define causality, so this aspect could be presented in the discussion, and it could also be considered the existence of third mediating variables.

  →Thank you for your comment. We added as follows.

・Page 8, line 211-212: The results were based on the relationships between the suicide rate and four factors and do not define causality.

  1. Conclusions

 The conclusion part should be further expanded, so the authors are invited to present identified limitations for the present research and possible insights for future research.

→Thank you for your comment. Reviewers 1 and 3 also showed about limitations. Based on precious comments of Reviewers 1,2 and 3, we added contents their limitations in Discussion.

References 

This session isn’t entirely in line with the formatting required by the journal guidelines, since the number volume of the journals should be in written in italics.  Overall, the authors are invited to make the reference more diverse in terms of sources, as a lot of the literature focuses on Japan and on literature produced by the same author (see, for instance, 7, 12, 13, 19-27). As regards suicide prevention interventions, some suggested literature the authors could refer to is:

→Thank you for your advice. We delated their almost references [previous Ref No7, 12,13, 19-22, 25 and 27] and sentences.

- Testoni, I., Tronca, E., Biancalani, G., Ronconi, L., & Calapai, G. (2020). Beyond the wall: death education at middle school as suicide prevention. International journal of environmental research and public health17(7), 2398. doi: 10.3390/ijerph17072398

→Thank you for your comment.

・We wrote number volume of the journals in italics.

・We added the reference, and we wrote the parts in Page 8, line 206-210.

  1. Testoni, I.; Tronca, E.; Biancalani, G.; Ronconi, L.; Calapai, G. Beyond the Wall: Death Education at Middle School as Suicide Prevention. Int. J. Environ. Res. Public Health 2020, 17, 2398.

It is necessary to perform further research on suicide in Kazakhstan. Suicide prevention is important from a young age, and prevention programs has been improved the ability to recognize and verbally convey emotions while maintaining stable initial characteristics such as psychological well-being and positive expectations for the future [41].

Use of English

 Overall, the authors use a correct and understandable English language. Some parts of the text should be further explained, as previously mentioned above.

 →A profession editing service (KN International) edited all of this revise manuscript.

Overview

 In general, the present research requires major revisions. The authors are invited to re-structure both the introduction and discussion sessions. They are invited to maintain only those pieces of information relevant to understand the background and the context in which this research is located. Moreover, more international literature should be used in the references for this article, and it should be avoided to use too many references from the same author.

→Thank you for your comment. We corrected this paper in all sections of Abstract, Keywords, Introduction, Materials and Methods, Results, Discussion, Conclusions, Figure, and Table based on Reviewers 1, 2 and 3.

Other

→・Reference No is new in this time.

Very sincerely yours,

Correspondence: Ken Inoue, MD, PhD

Research and Education Faculty, Medical Sciences Cluster, Health Service Center, Kochi University

The Center for Peace, Hiroshima University

Reviewer 3 Report

This is, in summary, an interesting study aimed to investigate labor, financial, and economic factors and whether any of these factors may contribute to the risk of suicide in Kazakhstan. The authors found that the unemployment rate was the most closely correlated with the suicide rate. Importantly, the suicide rate was associated only with the unemployment rate.

The authors may find as follows my main comments/suggestions.

First, when throughout the Introduction section, the authors correctly reported that substance use (injection drug use), intimate partner violence, and HIV on depression may be considered important risk factors for suicidal thoughts in Kazakhstan, they could even briefly mention the importance and impact of childhood traumatic experiences on suicidal behavior. Importantly, childhood traumatic experiences may be associated with psychosocial impairment and may influence negative outcomes. Thus, given the importance of this topic (although i understand that the link between traumatic experiences, and suicidal behavior is not the main topic of the present paper), i suggest to cite within the main text the article published on Frontiers in Psychiatry in 2017 (PMID: 28970807).

In addition, the involvement of sensory perception which is implicated in emotional processes and negative outcomes such as suicidality, might be briefly discussed. Importantly, the unique sensory processing patterns of individuals have been reported as crucial factors in determining negative outcomes in the clinical practice. Thus, given the above information, my suggestion is to include within the manuscript, the study published in 2016 on Child Abuse Negl (PMID: 27792883).

In addition, as the most relevant aims/objectives of this study have been well described, the main hypotheses underlying this study could be reported in a more detailed manner.

Importantly, how specifically labor, financial and economic items that are risk factors for suicide in Kazakhstan were revealed, as well as the exact way in which new indicators were identified need to be adequately in comprehensively described for the general readership.

Also, the main limitations/shortcomings might be provide in a more detailed manner as the description of the main caveats has been only partially provided for the general readership.  

Finally, what is the take-home message of this manuscript? While the authors reported that among several labor, financial, and economic factors, unemployment has been a major risk factor for suicide in the overall population, males, and females in Kazakhstan, they failed, in my opinion, to focus on some conclusive remarks to this specific regard. Specifically, which type of preventive measures and prevention campaigns need to be conducted in the real-world in order to effectively prevent suicidal behavior? Here, some additional information might be useful for the readers.

Author Response

Reviewer 3:

This is, in summary, an interesting study aimed to investigate labor, financial, and economic factors and whether any of these factors may contribute to the risk of suicide in Kazakhstan. The authors found that the unemployment rate was the most closely correlated with the suicide rate. Importantly, the suicide rate was associated only with the unemployment rate.

 →Thank you for your comment. We are glad.

The authors may find as follows my main comments/suggestions.

First, when throughout the Introduction section, the authors correctly reported that substance use (injection drug use), intimate partner violence, and HIV on depression may be considered important risk factors for suicidal thoughts in Kazakhstan, they could even briefly mention the importance and impact of childhood traumatic experiences on suicidal behavior. Importantly, childhood traumatic experiences may be associated with psychosocial impairment and may influence negative outcomes. Thus, given the importance of this topic (although i understand that the link between traumatic experiences, and suicidal behavior is not the main topic of the present paper), i suggest to cite within the main text the article published on Frontiers in Psychiatry in 2017 (PMID: 28970807).

→Thank you for your specific suggestion. We added as follows.

・Page 1, line 36-38: A recent systematic review of medical and social science literature regarding the relationship between childhood maltreatment and non-suicidal self-injury confirmed a direct relationship between the two [12].

  1. Serafini, G.; Canepa, G.; Adavastro, G.; Nebbia, J.; Belvederi Murri, M.; Erbuto, D.; Pocai, B.; Fiorillo, A.; Pompili, M.; Flouri, E.; Amore, M. The Relationship between Childhood Maltreatment and Non-Suicidal Self-Injury: A Systematic Review. Front. Psychiatry 2017, 8, 149.

In addition, the involvement of sensory perception which is implicated in emotional processes and negative outcomes such as suicidality, might be briefly discussed. Importantly, the unique sensory processing patterns of individuals have been reported as crucial factors in determining negative outcomes in the clinical practice. Thus, given the above information, my suggestion is to include within the manuscript, the study published in 2016 on Child Abuse Negl (PMID: 27792883).

→Thank you for your further specific advice. We added as follows.

・Page 1, line 38-41: Sensory perception is another factor implicated in emotional processes and suicidality, and researchers are coming to understand that sensory processing patterns related to childhood traumatic experiences may be important in the prediction of quality of life [13].

  1. Serafini, G.; Gonda, X.; Pompili, M.; Rihmer, Z.; Amore, M.; Engel-Yeger, B. The relationship between sensory processing patterns, alexithymia, traumatic childhood experiences, and quality of life among patients with unipolar and bipolar disorders. Child Abuse Negl. 2016, 62, 39-50. 

In addition, as the most relevant aims/objectives of this study have been well described, the main hypotheses underlying this study could be reported in a more detailed manner.

→Thank you for your comment. We thank to your specific suggestions.

Importantly, how specifically labor, financial and economic items that are risk factors for suicide in Kazakhstan were revealed, as well as the exact way in which new indicators were identified need to be adequately in comprehensively described for the general readership.

→Thank you for your comment. We added as follows.

・Page 2, line 75-77: Therefore, discussions of the influence on suicide of labor, financial, and economic factors in Kazakhstan can only be based on surmise, not actual data.

Also, the main limitations/shortcomings might be provide in a more detailed manner as the description of the main caveats has been only partially provided for the general readership.  

→Thank you for your valuable suggestion. Reviewer 1 was also shown description of limitations. We added as follows.

・Page 8, line 211-226: There are some limitations to this study. The results were based on the relationships between the suicide rate and four factors and do not define causality. Furthermore, the present study is a discussion about economic-based risk factors for suicide based on statistical analysis from only numerical data without individual information. Analyses of case-based studies of suicide would be important from the viewpoint of clinical and societal interventions. The study does not examine differences between Soviet-era and post-Soviet suicide rates, so it cannot shed light on the effect of a command economy vs market economy on suicide. The study does not probe the stark difference between male and female suicide rates in Kazakhstan or compare them to those of other countries and cultures where masculinity may be defined differently. The study does not examine differences between conditions in countries like Japan and the U.S., where unemployment is related, and in Taiwan and Sweden, where it is not related. It does not elaborate what types of prevention programs have worked or examine the data for clues to whether those methods might be applicable to Kazakhstan. However, it is a timely exploration given the current economic conditions worldwide and opens up these issues as opportunities for further study among researchers in a variety of academic fields.

Finally, what is the take-home message of this manuscript? While the authors reported that among several labor, financial, and economic factors, unemployment has been a major risk factor for suicide in the overall population, males, and females in Kazakhstan, they failed, in my opinion, to focus on some conclusive remarks to this specific regard. Specifically, which type of preventive measures and prevention campaigns need to be conducted in the real-world in order to effectively prevent suicidal behavior? Here, some additional information might be useful for the readers.

→Thank you for your valuable suggestion. Reviewer 1 was also shown similar suggestion to you. Therefore, we added as follows.

・Page 8, line 231-236: As prevention measures, it is important for labor, financial, and economic divisions of government at both the local and national level to collaborate in planning for economic downturns and provide stabilizing supports such as jobs programs or organized volunteerism to maintain individuals’ sense of self and purpose. In addition, extra efforts to implement suicide awareness and prevention programs at schools, community centers and places of worship during times of economic hardship are called for.

Other

→・Reference No is new in this time.

・A profession editing service (KN International) edited all of this revise manuscript.

Very sincerely yours,

Correspondence: Ken Inoue, MD, PhD

Research and Education Faculty, Medical Sciences Cluster, Health Service Center, Kochi University

The Center for Peace, Hiroshima University

Round 2

Reviewer 2 Report

Dear Authors,

your work now is really accurate and interesting. Congratulations on the results.

Sincerely

Reviewer 3 Report

In the revised manuscript, the authors addressed most of the major questions raised by Reviewers improving both the main structure and quality of the present paper. I have no further additional comments.